# Design and Synthesis of Novel *N*-Benzylidene Derivatives of 3-Amino-4-imino-3,5-dihydro-4*H*-chromeno[2,3-*d*]pyrimidine under Microwave, *In Silico* ADME Predictions, *In Vitro* Antitumoral Activities and *In Vivo* Toxicity

**DOI:** 10.3390/ph17040458

**Published:** 2024-04-02

**Authors:** Sirine Karoui, Marwa Dhiabi, Mehdi Fakhfakh, Souhir Abid, Emmanuelle Limanton, Rémy Le Guével, Thierry D. Charlier, Anthony Mainguy, Olivier Mignen, Ludovic Paquin, Houcine Ammar, Jean-Pierre Bazureau

**Affiliations:** 1Laboratoire de Chimie Appliquée: Hétérocycles, Corps et Polymères, Faculté des Sciences de Sfax, Université de Sfax, Route Soukra, BP 1171, Sfax 3000, Tunisia; sirinekaroui147@gmail.com (S.K.); dhiabi.marwa1@gmail.com (M.D.); mehdi.fakhfakh@gmail.com (M.F.); houcine.ammar@fss.rnu.tn (H.A.); 2Institut des Sciences Chimiques de Rennes, ISCR UMR CNRS 6226, Université de Rennes, Campus de Beaulieu, Bât. 10A, 263 Avenue du Général Leclerc, CS 74205, 35042 Rennes CEDEX, France; ludovic.paquin@univ-rennes.fr; 3Department of Chemistry (Science and Arts): Al Qurayat, Al-Jouf University, Al-Qurayyat P.O. Box 756, Al Jawf, Saudi Arabia; souhirabid74@gmail.com; 4S2Wave Platform, ScanMAT UAR 2025 CNRS, Université de Rennes, Campus de Beaulieu, Bât. 10A, 263 Avenue du Général Leclerc, CS 74205, 35042 Rennes CEDEX, France; emmanuelle.limanton@univ.rennes.fr; 5ImPACcell Platform, Biosit, SFR UMS CNRS 3480, Inserm 018, Campus de Villejean, Bât. 8, 2 Avenue du Prof. Léon Bernard, 35043 Rennes, France; remy.leguevel@univ-rennes.fr (R.L.G.); thierry.charlier@univ-rennes.fr (T.D.C.); 6Institut de Recherche en Santé, Environnement et Travail, IRSET Inserm UMR_S 1085, 9 Avenue du Prof. Léon Bernard, 35000 Rennes, France; 7Lymphocytes B & Auto Immunité, LBAI Inserm UMR 1227, Université Bretagne Occidentale, 29 Avenue Camille Desmoulins, 29200 Brest, France; anthony.mainguy@univ-brest.fr (A.M.); olivier.mignen@univ-brest.fr (O.M.)

**Keywords:** 2-amino-4*H*-chromene, 4*H*-chromeno[2,3-*d*]pyrimidine, microwave irradiation, tumoral cell line, cytotoxicity, Lipinski’s rules of five, *in silico* ADME predictions, embryo toxicity, zebrafish *Danio rerio*

## Abstract

The synthesis of a series of new *N*-benzylidene derivatives of 3-amino-4-imino-3,5-dihydro-4*H*-chromeno[2,3-*d*]pyrimidine **10(a-l)** bearing two points of molecular diversity is reported. These new compounds were synthesized in five steps including two steps under microwave dielectric heating. They were fully characterized using ^1^H and ^13^C NMR, FTIR and HRMS. The *in silico* physicochemical properties of compounds **10(a-l)** were determined according to Lipinski’s rules of five (RO5) associated with the prediction of their bioavailability. These new compounds **10(a-l)** were tested for their antiproliferative activities in fibroblasts and eight representative human tumoral cell lines (Huh7 D12, Caco2, MDA-MB231, MDA-MB468, HCT116, PC3, MCF7 and PANC1). Among them, the compounds **10h** and **10i** showed sub-micromolar cytotoxic activity on tumor cell lines (0.23 < IC_50_ < 0.3 μM) and no toxicity on fibroblasts (IC_50_ > 25 μM). A dose-dependent inhibition of Store-Operated Ca^+2^ Entry (SOCE) was observed in the HEK293 cell line with **10h**. *In vitro* embryotoxicity and angiogenesis on the mCherry transgenic zebrafish line showed that **10h** presented no toxic effect and no angiogenic effect on embryos with a dose of 5 μM at 72 hpf.

## 1. Introduction

Over the two last decades, the 2-amino-4*H*-chromene and their 4*H*-chromeno[2,3-*d*]pyrimidine derivatives have been identified as promising small molecule targets in medicinal chemistry due to the fact that these compounds exhibited a wide range of biological activities (Figure 1). In the series of 2-amino-4*H*-chromene compounds, *Crolibulin*^TM^ (EPC2407) or (4*R*) 2,7,8-triamino-4-(3-bromo-4,5-dimethoxyphenyl)-4*H*-chromene-3-carbonitrile **A** were evaluated in 2016 in phase I/II clinical trials for the treatment of aggressive solid tumors [1]. During the randomized phase II clinical trial, the *Crolibulin*^TM^ drug candidate was combined with cisplatin and compared to cisplatin alone on a panel of 40 enrolled patients with a focus on anaplastic thyroid cancer (NTC 01240590) [2]. The ethyl 2-amino-6-bromo-4-(1-cyano-2-ethoxy-2-oxoethyl)-4*H*-chromene-3-carboxylate named HA14-1 **B** [3,4,5] as another small molecule, was identified as a potential non-peptidic Bcl-2 inhibitor by using an *in silico* drug design strategy based on the predictive structure of Bcl-2 protein. MX 58151 **C** or 2-amino-4-(3-bromo-4,5-dimethoxyphenyl)-4*H*-chromene-3-carbonitrile, a similar compound to HA 14-1 **B**, appeared as tubulin inhibitor [6] with a GI_50_ of 37 nM in T47D breast cancer cells [7]. For SV30 **E** or ethyl (2-amino-6-bromo-4-diethylaminomalonate)-4*H*-chromene-3-carboxylate, it was combined with lipid nano-capsules (LNCs) on F98 tumoral cells to improve its biological activities [8]. For naphthapyran LY290181 **E**, or (2-amino-4-pyridyl)-4*H*-naphtho[1,2-*b*]pyran-3-carbonitrile, this one appeared in 1997 as a potent antiproliferative compound blocking cells in the G_2_/M phase of the cell cycle associated to action on microtubules [9,10].

Now for the chromeno[2,3-*d*]pyrimidine derivatives (Figure 1), the 3-(benzylidenamino)-5-(4-bromophenyl)-4-imino-4,5-dihydro-3*H*-chromeno[2,3-*d*]pyrimidine-8-ol **F** presented cytotoxic effects on MCF7, HCT-116, HepG-2 and A549 cancer cells [11]. It is interesting to note that the 5-(4-chlorophenyl)-*N^8^*,*N^8^*-diethyl-4-amino-4*H*-chromeno[2,3-*d*]pyrimidine-3,8(5*H*)-diamine **G** was also identified as an antitumoral compound on MCF7 (human breast tumor cells) and HCT-116 (human lung tumor cells) [12] associated with urease inhibition activity [13]. The introduction of a 1,2,4-triazolo moiety on the chromeno[2,3-*d*]pyrimidine skeleton of compound **H** does not affect the cytotoxic effect on MCF7 tumor cell lines [14] in comparison to compound **G**. A novel series of dialkyl (4-amino-5*H*-chromeno[2,3-*d*]pyrimidin-5-yl)phosphonates were synthesized by Reddy’s team that inhibits the growth of A549, DN-145, PC3, HeLa and MCF7 tumor cell lines through in vitro MTT assays, and among them, the dimethyl (4-amino-5*H*-chromeno[2,3-*d*]pyrimidin-5-yl)phosphonate **I** was identified as a promising antitumor agent [15].

In this context and due to the synthetic potential around the 2-amino-4*H*-chromene platform and their chromeno[2,3-*d*]pyrimidine derivatives, the herein study was designed in order to identify novel potential anticancer compounds. In continuation of previous work [16], it seemed interesting to synthesize some new *N*-benzylidene derivatives of 3-amino-4-imino-3,5-dihydro-4*H*-chromene[2,3-*d*]pyrimidines and to appraise their cytotoxic activities on a panel of tumoral cell lines. To better understand the pharmacological interest of these novel compounds, ADME predictions will be investigated. To complete the biological profile of the most active compounds on tumor cells, their effects will be evaluated on store-operated-calcium entry (SOCE) as this calcium entry is implicated in several human cancers [17]. Owing to the fact that the zebrafish *Danio rerio* embryo model is accepted as a validated alternative assay to assess the acute toxicity of a molecule on this fish (OECD, N° 236) [18], the embryo toxicity evaluation will be examined with the selected relevant antitumor compounds and also associated with their possible effect on angiogenesis in the zebrafish embryo.

## 2. Results and Discussion

### 2.1. Synthesis Chemistry

In view of the relevance of *N*-arylidene derivatives of 3-amino-4-imino-3,5-dihydro-4*H*-chromene[2,3-*d*]pyrimidine as potential antitumor agents, the retrosynthesis analysis (Figure 2) showed that the desired starting compounds included three major building-blocks. The first is the 2-amino-4*H*-chromene-3-carbonitrile platform, and the second is hydrazine, which provides the link between the platform and the *N*-arylidene moiety after condensation with an aromatic aldehyde. In this type of structure, the molecular diversity should come from the starting 2-amino-4*H*-chromene platform and aromatic aldehydes.

For the construction of desired 3-(*N*-phenylmethyliden)amino-4-imino-3,5-dihydro-4*H*-chromeno[2,3-*d*]pyrimidine, the synthetic route of targeted compound **10** is depicted in Figure 1. Concerning the 2-amino-4*H*-chromene-3-carbonitrile **4** as a key intermediate in this synthetic strategy, the preparation involved a two-step approach initially developed in our laboratory [19] and well mastered practically in scale-up. In the first step for the synthesis of 2-imino coumarins **3(a-d)**, the first point of molecular diversity was introduced by using four various aromatic aldehydes **1(a-d),** which are, respectively, **1a**, 4-(diethylamino)salicylaldehyde; **1b**, salicylaldehyde; **1c**, 2-hydroxy-3-methoxybenzaldehyde or ortho vanillin; and **1d**, 2-hydroxynaphthaldehyde. Access to 2-imino coumarins **3** is very simple: reaction of commercial aldehyde **1(a-d)** with propanedinitrile **2**, in ethanolic solution at room temperature, in the presence of piperidine 0.5% mol. produced the desired compounds **3** after work-up in good yields. In the second step, the preparation of the 2-amino-3*H*-chromene-3-carbonitriles **4** resulted from the reduction of 2-imino coumarins **3** with sodium borohydride at 0 °C in a cooled solution of MeOH during 45 min. Due to the low solubility of these compounds **4** in MeOH, they were recovered using simple filtration on sintered glass (porosity N° 4).

With the underlying idea to introduce the pyrimidine fragment on 4*H*-chromene skeleton, it is necessary to add an electrophilic carbon to the 2-amino function of 4*H*-chromene-3-carbonitrile **4** for hetero cyclization. For this, the transformation of the 2-amino function of **4** into imidate **6** works well by the addition of 6 equivalents of ethyl orthoformate **5** at 110 °C under microwave dielectric heating [20] for 60 min catalyzed with glacial acetic acid 0.5% mol., which afforded the desired functionalized imidate **6** (Table 1) in moderate (**6c**: 52%) to good yields (**6d**: 90%). The interest of this method is that compound **6** is not soluble in the reaction mixture, which facilitates recovery using simple filtration. For the aza-annulation of imidate **6** into 4-imino-4,5-dihydro-3*H*-chromeno[2,3-*d*]pyrimidine **8**, we applied a method described in the literature [21] employing the hydrazine reagent **7**. Applying this approach with slight experimental modifications, we treated imidate **6** with 1.6 equivalents of aqueous solution of hydrazine (35% *w*/*v*) at room temperature in EtOH for 2 h. As the reaction progressed, we noticed the low solubility of 3-amino-4-imino-3,5-dihydro-4*H*-chromeno[2,3-*d*]pyrimidine **8** in the reaction media which, again, facilitates their recovery by simple filtration. These compounds **8(a-d)** were prepared in yields ranging from 50 to 90%. If necessary, these intermediates **8** can be easily purified by recrystallization from EtOH. For the fifth and last step, the introduction of the second point of molecular diversity was realized by the condensation of 3-amino compound **8** with aromatic aldehydes **9** or **1**. After a set of experiments, optimal reaction conditions for access of the desired 3-(*N*-phenylmethyliden)amino-4-imino-3,5-dihydro-4*H*-chromeno[2,3-*d*]pyrimidine **10(a-i)** and 10-(*N*-phenylmethyliden)amino-11-imino-11*H*-naphto[1′,2′:5,6]pyrano[2,3*-d*]pyrimidine **10(j-l)** were obtained from a mixture of aromatic aldehyde **9** and compound **8** in absolute EtOH using a reaction temperature of 160 °C for 60 min under microwave irradiation in the presence of piperidine 0.5% mol. After cooling down, filtration of the crude reaction mixture afforded compound **10,** which was purified using recrystallization from EtOH (**10**: 26–90% yield). The structure assignment of compound **10** was evidenced by FTIR (C=N stretch, 1615 < ν < 1645 cm^−1^), by ^1^H NMR for N-N=CH signal (8.24 < ν < 8.59 ppm) which is in agreement with an *E*-configuration [22]. In mass spectrometry analysis (HRMS), the [M+H]^+^ molecular ion signals for all compounds **10(a-l)** were readily obtained as base signals.

### 2.2. Physicochemical Properties of Compounds ***10(a-l)***

Lipinski’s benchmark rule of five (RO5) defines desirable drug candidate physicochemical properties, such as the molecular weight MW < 500 Da, the octanol/water partition coefficient log P_o/w_ < 5, the number of hydrogen bond donor HBD < 5, the number of hydrogen bond acceptor HBA < 10, the rotatable bonds RB < 10 [23] and completed now with the topological polar surface area *t*PSA < 140 Å^2^ [24]. The conventional RO5 has gained popularity as a “good rule of thumb” but is not specific for all drugs, especially for central nervous system (CNS) drugs [25]. In the actual context, the physicochemical properties of compounds **10(a-l)** are easily accessed on the SwissADME [26] web server [27], and the results are summarized in Table 2.

For compound **10**, the log P_o/w_ values are acceptable (2.86 < log P_o/w_ < 4.07). The highest value (4.07) corresponds to compound **10a** and the lowest values (2.66 and 2.86) correspond to compounds **10c** and **10h** bearing, not surprisingly, an ionizable phenolic function. For predicting drug solubility, the *t*PSA (topological polar surface area) is another interesting and effective descriptor for the transport properties of a molecule. The *t*PSA parameter describes the sum of the surface of its polar atoms (N, O) and the hydrogen atoms attached to them and, it is an interacting indicator of gastrointestinal absorption [28]. For compounds **10(a-l),** all of them have *t*PSA < 140 Å^2^ (63.26 < *t*PSA**_10(a-l)_** < 92.72 Å^2^) and evidenced an acceptable permeability. To complete this, the calculated values of aqueous solubility (log S) showed that most of compound **10** is moderately soluble in water (−5.5 < log S**_10(a-l)_** < −3.9). If we take into account the other parameters, such as hydrogen bond acceptor HBA (HBA**_10(a-l)_** < 5), hydrogen bond donor HBD (HBD**_10(a-l)_** < 2), and rotatable bonds RB (2 < RB**_10(a-l)_** < 6), we note that all of compound **10** is in agreement with Lipinski’s rule of five (RO5) (LV**_10(a-l)_** = 0) [23]. This means that these compounds should have potential efficient absorption and good oral bioavailability.

### 2.3. Cytotoxicity Assays of Compounds ***10(a-l)***

The potential antiproliferative activity of these novel 3-(*N*-phenylmethyliden)amino-4-imino-3,5-dihydro-4*H*-chromeno[2,3-*d*]pyrimidine **10(a-i)** and 10-(*N*-phenylmethyliden)amino-11-imino-11*H*-naphto[1′,2′:5,6]pyrano[2,3*-d*]pyrimidine **10(j-l)** was evaluated in a panel of eight tumoral cell lines and in diploid skin fibroblasts used as normal cell lines for control. For these antitumor assays, chosen cell lines represent diverse types of human cancer, which are, respectively, human hepatocellular carcinoma (Huh7 D12), human colon adenocarcinoma (Caco2), human breast adenocarcinoma (MDA-MB231), MDA-MB468 for human breast carcinoma which correspond to the Triple Negative Breast Cancer (TNBC), human colon carcinoma (HCT116), PC3 as cell line initiated from bone metastasis of grade IV prostatic adenocarcinoma from a 62-year-old white man, MCF7 as breast cancer line isolated from a 69-year-old white woman and PANC1, as human pancreatic cancer cell line. The potential antiproliferative activity was preliminarily evaluated in vitro at a 25 μM single dose. The roscovitin and doxorubicin compounds were used in all experiments as reference standards because they are wide-spectrum anticancer agents.

For the compounds exhibiting strong growth inhibitory activity against a tumoral cell line (with a percentage of survival is <50%), they were used for a more detailed study to obtain full dose-response and survival curves using different concentrations (0.1–25 μM) for the determination of IC_50,_ which represented the concentration of the compound that kills 50% of the tumoral cells after 48 h incubation. The IC_50_ was calculated from the full dose-response curve and used as a parameter for the antiproliferative activity of the new compounds. The results of these cytotoxicity assays for compounds **10(a-i)** and compounds **10(j-l)** are reported in Table 3.

The analysis of the results in Table 3 for in vitro antiproliferative activity assays highlights three categories of compounds. The first category concerns inactive products with IC_50_ > 25 μM on tumor cancer cell lines and on fibroblasts. This concerns compounds **10(a-c)**, 1**0f** and **10l**. The second category is for compound **10** presenting moderate activity on tumor cell lines (1 < IC_50_ < 25 μM) but is cytotoxic on fibroblasts (healthy skin cell); this is the case for compounds **10(d, e)**, **10g** and **10(j, k)**. The last and the most interesting category concerns compounds with sub-micromolar cytotoxic activity on tumor cell lines (IC_50_ < 1 μM) but without an effect on fibroblasts. For this most interesting category, compounds **10h** and **10i** (Figure 3) are in agreement with our selection criteria, i.e., a molecule active on tumor cells (IC_50_ < 1 μM) and inactive on non-cancer cells (fibroblasts, IC_50_ > 25 μM).

### 2.4. In Silico ADME Properties of Compounds ***10h*** and ***10i***

To complete the physicochemical properties of compound **10** according to the Lipinski rule of five (RO5), we decided to predict for our compounds their gastrointestinal absorption (HIA-human gastrointestinal absorption) and BBB (Blood–Brain Barrier) penetration by using the *W*-Log P-*versus*-*t*PSA referential named “BOILED-Egg” model (Figure 4A) obtained from the SwissADME server [26]. The “BOILED-Egg” model provides a quick, simple, easily repeatable, but statically unparalleled robust strategy for forecasting the passive gastrointestinal absorption and brain access of tiny compounds relevant for drug discovery and development. In this model, the “white region” is for the high probability of passive absorption by the gastrointestinal tract, and the “yellow region” (yolk) is for the high probability of brain penetration. Yolk and white areas are not mutually exclusive. The “gray region” concerns compounds that are predicted not to be absorbed in the gastrointestinal tract and not BBB permeant. The molecules are represented inside as either blue, which are substrates of P-glycoprotein, or red, which are not. In the BOILED-Egg model, we can see that compound **10h** is located in the white region of high probability for passive gastrointestinal absorption and is not effluxed by P-glycoprotein (red circle). On the contrary, compound **10i** is in the yellow area which means a high probability of passive BBB penetration and is also not effluxed by P-glycoprotein. This in silico result of the difference of BBB penetration between **10h** (log P**_10h_** 2.86) and **10i** (log P**_10i_** 3.17) is not surprising because **10i** does not have a phenolic function (polar and ionizable group) [29]. For the other compounds bearing a hydroxyl group in this series, which concerns **10b**, **10e** and **10k**, the results are reported in Appendix A.

For the bio-availability radar of compounds **10(h, i)** in Figure 4B, which is the correlation of the six factors, LIPO (Lipophilicity): −0.7 < Log P < +5.0; SIZE: 150 g/mol < MW < 500 g/mol; POLAR (Polarity): 20 Å^2^ < *t*PSA < 130 Å^2^; INSOLU (Insolubility): −6 < Log S < 0; INSATU (Unsaturation): 0.25 < Fraction Csp^3^ < 1; FLEX (Flexibility): 0 < Number rotatable bonds < 9, we can see that molecules **10(h, i)** violated just one criterion, i.e., unsaturation because both have fraction Csp^3^ parameter < 0.25 (Table 2). The pink-colored zone is the suitable physicochemical space for oral bioavailability and results for the other compounds **10(a-l)** are summarized in Appendix A. For in silico prediction of drug-likeness model (Figure 4C), compound **10h** presents, respectively, a score of −0.45 for **10h** and −0.55 for **10i** [30]. In the analysis of all these in silico predictions and from a point of view of pharmacology, the most interesting compound **10** is molecule **10h** because this one does not present a passive BBB penetration, this is the case of **10i,** which is not effluxed by P-glycoprotein.

### 2.5. Store-Operated Calcium Entry (SOCE) Assays for Compound ***10h***

Store-operated Ca^+2^ entry (SOCE) is a major pathway of Ca^+2^ entry in mammalian cells, which regulates a large variety of cellular functions including proliferation, apoptosis, motility and death. The SOCE is a ubiquitous receptor-activated Ca^+2^ entry route across the plasma membrane that is critical to many cellular functions in mammalian cells [31]. Aberrant SOCE is directly related to carcinogenesis, cancer cell proliferation, chemoresistance, angiogenesis and metastasis [32,33]. SOCE is supported mainly by the plasma membrane (PM) calcium selective channels of the Orai family and regulated by the Ca^+2^ sensing stromal interaction molecules (STIM1 and STIM2) located in the membrane of the endoplasmic reticulum (ER).

Today, it is well established that the depletion of ER Ca^+2^ induces STIM1 conformational changes and its translocation to ER-plasma membrane junctions where they activate Orai channels supporting SOCE [34,35]. In many cancers, such as breast, liver, lung, gastric and colon, accumulating evidence exhibits augmented SOCE and modification of STIM1 and Orai protein expression. SOCE inhibition decreases the proliferation and metastasis of cancer cells, suggesting that SOCE may act as an oncogenic pathway [36]. Also, the need and the challenge of discovering small molecules modulating SOCE [37] will be very important in cancer with the underlying idea to shoot out the uncontrolled problem of metastasis [38].

Regarding the fact that compound **10h** or (*3E*) 3-*N*-(2-hydroxyphenylmethylidene)amino-4-imino-9-methoxy-3,5-dihydro-4*H*-chromeno[2,3-*d*]pyrimidine presents noticeable sub-micromolar cytotoxic activity on several tumor cell lines (**10h**: 0.23 < IC_50_ < 0.3 μM on Huh7 D12, Caco2, MDA MB-231, MDA MB-468, HCT116, PC3 and MCF7) without toxicity on fibroblasts (IC_50_ > 25 μM), we evaluated initially its inhibitory effect on SOCE in HEK293 cell lines (Human Embryonic Kidney). Thapsigargin (TG), an inhibitor of endoplasmic reticulum Ca^+2^-ATPase, is used to deplete intracellular calcium stores in a Ca^+2^-free solution and therefore activate SOCE. After the addition of 2 μM TG in a Ca^+2^-free solution, HEK293 cells exhibited a rapid rise of intracellular Ca^2+^ concentration ([Ca^2+^]_i_) (Figure 5A) corresponding to ER Ca^2+^ stores depletion. Subsequent reintroduction of 1.8 mM CaCl_2_ to the extracellular solution resulted in a sustained increase in [Ca^2+^]_i_ from baseline, which is consistent with a characteristic SOCE-mediated Ca^+2^ influx from the extracellular solution. Pre-treatment with compound **10h** at a concentration ranging from 1.25 to 50 μM (addition after 5 min before recording iCa^+2^ level variation) partially inhibits SOCE in HEK293 cells in a dose-dependent manner (10–50 μM) (Figure 5B for % release and Figure 5C for % SOCE). The evaluation of the same compound **10h** on a B cell line (PLP cell line) with the same protocol, presented no inhibitory effects on SOCE and Ca^+2^ release. In the context of a preliminary exploration of the functional effects of our molecules of interest and in the absence of a significant inhibition at a low concentration of compound **10h** on the reference cell line we used (HEK293) no further exploration of the modulatory effect of this compound on cancer cell lines was initiated.

### 2.6. In Vitro Embryotoxicity and Angiogenesis of Compound ***10h*** on Zebrafish Danio rerio

Zebrafish embryos are increasingly used as in vivo models for drug screening, drug toxicity and safety assessments due to their original advantages. Zebrafish have orthologues for 70% of human proteins and comparable vertebrate anatomy, as well as paralogues of 84% of all known disease-related genes [39]. The optical transparency of the zebrafish embryos makes it easy to visualize directly the changes in the developing vascular system [38]. Since 2013, the zebrafish embryo has been accepted as a validated model for alternative assay to assess fish acute toxicity (OECD, N° 263) [18]. Another interest in the zebrafish embryo model relies on the fact that the effects are evaluated on the entire vertebrate organism during the period of organogenesis. To evaluate in vivo embryotoxicity of compound **10h** on zebrafish, we used the Tg(kdrl:Has.HRAS-mcherry)^s896^ zebrafish transgenic model and, sorafenib and erlotinib as reference drugs for comparison. Sorafenib is a multi-kinase inhibitor that reduces and decreases tumor angiogenesis and tumor cell proliferation in vivo inhibiting tumor growth of a broad spectrum of human tumor xenographs in athymic mice. Erlotinib is an anticancer drug and a potent inhibitor of intracellular EGRF phosphorylation.

The embryos were exposed to the test compounds (**10h**, sorafenib and erlotinb) according to standard protocols described in the literature [18,40]. Fertilized eggs at 24–48 h post-fertilization (hpf) were selected and transferred to a 96-well plate (1 embryo/well). Concentrations used range from 0.021 to 5 μM and for compound **10h** and sorafenib and 0.01 to 3 μM for erlotinib). Drugs were added to the respective well and the plate was then incubated at 28.5 °C. DMSO (0.6%) was used as vehicle control for the test compounds. The exposure duration continued up to 72 hpf because it is the endpoint of embryonic development [41]. The dead embryos were eliminated, and the fertilized eggs were imaged at 0, 24, 48 and 72 hpf. The morphological deformities of zebrafish embryos were examined using an inverted Zeiss microscope, equipped with a monochromatic sCMOS camera (HAMAMATSU ORCA FLASH 4 LT) and a Colibri illumination system, controlled by Zen software 3.7 (Zeiss, Jena, Germany).

Representative images of the embryos (magnification: 2.5×) after exposure to compound **10h** and sorafenib are presented in Figure 6, and in Appendix A for erlotinib. For low doses of sorafenib, the normal development of the embryos is observed at 24 hpf whereas the morphological development is delayed at 48 hpf with an apparent toxicity at the 5 μM dose. Finally, the toxicity (4/5 of the embryos) is apparent at 72 hpf in embryos treated with 1.67 μM with an absence of angiogenesis of the vessels. Erlotinib treatment for 48 and 72 hpf, a complete hatching of the embryos is observed with a delay in vessel angiogenesis at the 0.38 μM dose (no toxicity at 1 μM dose for 72 hpf in agreement with a complete hatching of the embryos). Finally, 48 and 72 hpf treatment with our compound **10h**, induces no toxic effect and results in complete hatching of the embryos without delay in vessel angiogenesis up to 5 μM.

Another interest of zebrafish *Danio rerio* is the possibility to deeply evaluate the effects of angiogenic compounds. In our model, the formation of most vessels is complete, the circulation is present, and the heart is beating after 28 hpf of development. Angiogenesis begins in the somites and in the anterior portion of the embryos. Many zebrafish blood vessels form by angiogenic sprouting and appear to require the same proteins required for blood vessel growth in mammals. By using the mCherry line transgenic zebrafish, it is possible to analyze the direct effect of a compound on the angiogenesis of dorsal vessels. Zebrafish embryos can survive up to a week without an intact circulatory system because defective vessel formation has no immediate lethal effect [42].

Images of dorsal vessels of mCherry Tg zebrafish exposed to compound **10h**, sorafenib and erlotinib for 72 hpf are presented in Figure 7 (Zeiss microscope; magnification: 10×). Sorafenib at 1.67 μM induces an inhibition of the dorsal vessel vascularization confirming that this drug has an angiogenic effect. With erlotinib, the angiogenic effect appears at a 3.3 μM dose. For the compound **10h**, we cannot observe an anti-angiogenic effect even with a 5 μM dose compared to untreated zebrafish embryos.

## 3. Conclusions

Nine 3-(*N*-phenylmethylidene)amino-4-imino-3,5-dihydro-4*H*-chromeno[2,3*-d*]pyrimidine **10(a-i)** and three 10-*N*-(phenylmethylidene)amino-11-imino-11*H*-naphto[1′,2′:5,6]pyrano[2,3*-d*]pyrimidine **10(j-l)** were designed, synthesized in five steps including two steps under microwave irradiation. Compounds were evaluated *in silico* according to Lipinski’s rules of five (RO5) and tested experimentally for their antiproliferative activities on a panel of eight tumoral cell lines associated with diploid skin fibroblasts as normal cell lines. The antitumoral assays revealed that compounds **10(h, i)** presents submicrolar cytotoxic activity (0.23 < IC_50_ < 0.8 μM) on tumor cell lines Huh7 D12, Caco2, MDA MB231, MDA MB468, HCT116, PC3 and MCF7. These two compounds have been assessed *in silico* for their ADME profile using the “BOILED-Egg” model and also for their bioavailability. Store-operated Calcium Entry (SOCE) assays of **10h** in HEK293 cell lines revealed a dose-dependent effect but with a minor inhibitory effect on SOCE at low concentrations inducing toxicity in cancer cell lines. *In vitro* embryotoxicity of compound **10h** was performed on mCherry transgenic zebrafish line and compared to sorefenib and erlotinib. With a 5 μM dose of **10h** at 72 hpf, complete hatching of the zebrafish embryos without delay in vessel angiogenesis is observed and without toxic effect. The molecule **10h** can be considered as a potential bioactive “hit” compound for cancer and now additional work is in progress in our laboratory to complete its biological profile.

## 4. Materials and Methods

### 4.1. Chemistry Section—General Remarks

All reagents and solvents were purchased from Acros Fisher Scientific (Illkirch, France) and Merck Sigma-Aldrich (Saint-Quentin-Fallavier, France) were used without further purification. Thin-layer chromatography (TLC) was accomplished on 0.2-mm precoated plates of silica gel 60 F-254 (Merck, Rahway, NJ, USA) with appropriate eluent. Visualization was made with ultraviolet light (254 and 365 nm) or with a fluorescence indicator. Infrared (IR) spectra were recorded on a Jasco FT-IR 420 spectrophotometer apparatus (Jasco France, Lisses, France) using potassium bromide pellets. ^1^H NMR spectra were recorded on BRUKER AC 300 Avance III (300 MHz) spectrometer, ^13^C NMR spectra on BRUKER AC 300 Avance III (75 MHz) spectrometer using DMSO-*d_6_* with tetramethylsilane as an internal reference. Chemical shifts are expressed in parts per million downfield. Data are given in the following order: δ value, multiplicity (s, singlet; d, doublet; t, triplet; q, quartet; quint: quintuplet, m, multiplet; br, broad), number of protons, coupling constants *J* is given in Hertz. The high-resolution mass spectra (HRMS) were recorded in positive mode using direct Electrospray infusion (ESI), respectively, on a Waters Q-Tof 2 or on a Thermo Fisher Scientific Q-Exactive spectrometers at the “Centre Régional de Mesures Physiques de l’Ouest” CRMPO ScanMAT (CRMPO platform, ScanMAT UAR 2025 CNRS, Rennes, France). Melting points were determined on a Kofler melting point apparatus and were uncorrected. Reactions under microwave irradiation (S2Wave platform ScanMAT UAR 2025 CNRS, Rennes) were realized in the Anton Paar Monowave 300^®^ microwave reactor (Anton Paar France, Les Ulis, France) using borosilicate glass vials of 10 mL equipped with snap caps (at the end of the irradiation, cooling reaction was realized by compressed air). The microwave instrument consists of a continuous focused microwave power output from 0 to 800W. All the experiments were performed using the stirring option. The target temperature was reached with a ramp of 3 min and the chosen microwave power stayed constant to hold the mixture at this temperature. The reaction temperature is monitored using a calibrated infrared sensor and the reaction time includes the ramp period. The microwave irradiation parameters (power and temperature) were monitored by the Monowave software package for the Anton Paar Monowave 300^®^ reactor. Solvents were evaporated with a BUCHI rotary evaporator (Villebon sur Yvette, France).

Compounds as 2-imino coumarin-3-carbonitrile **3(a-d)** and 2-amino-4*H*-chromene-3-carbonitrile **4(a-d)** were synthesized according to methods previously developed in our laboratory and described in the literature [19,20].

#### 4.1.1. Standard Procedure for Synthesis of Ethyl *N*-(3-Cyano-4*H*-chromen-2-yl)formimidate **6(a-c)** and Ethyl *N*-(2-Cyano-1*H*-naphto[2,1-*b*]pyran-3-yl)formimidate **6d** in the Monowave 300^®^ Microwave Reactor

In a cylindrical quartz reactor (Ø = 1.8 cm, 20 mL) 2-amino-4*H*-1-chromene-3-carbonitrile **4** (2.5 mmol), commercial ethyl orthoformate (**5**: 1.59 g, 1.64 mL, 15 mmol., 6 equiv.), glacial acetic acid AcOH (0.5% mol.) and a magnetic stirring bar were introduced successively. The glass tube was sealed with a snap cap and placed in the Monowave 300^®^ microwave cavity (P = 800 Watt). The stirred mixture was irradiated at 100 °C (after a ramp of 3 min from 20 to 100 °C) for 60 min. After microwave dielectric heating, the crude reaction mixture was allowed to cool down at room temperature for 30–45 min. The resulting insoluble product **6** was collected using filtration on a Büchner funnel (porosity N° 4), washed carefully with diethyl ether Et_2_O (2 × 10 mL), then dried under high vacuum (10^−2^ Torr) at room temperature for 60 min. The desired product **6** was further used without purification.

*Ethyl N-(3-cyano-7-diethylamino-4H-chromen-2-yl)formimidate* (**6a**). According to the standard procedure, **6a** was synthesized from 2-amino-7-diethylamino-4*H*-1-chromene-3-carbonitrile **4a** (2.5 mmol) in 60% yield as yellow powder; mp = 110–112 °C. IR (KBr, ν, cm^−1^): 1540 (C=C); 2212 (CN); 1647 (C=N). ^1^H NMR (DMSO-*d_6_*) δ: 1.07 (t, 6H, *J* = 6.9 Hz, 2-CH_3_); 1.32 (t, 3H, *J =* 7.1 Hz, CH_3_); 3.30 (q, 4H, *J* = 6.9 Hz, 2-CH_2_); 3.50 (s, 2H, CH_2_); 4.32 (q, 2H, *J* = 7.1 Hz, O-CH_2_); 6.33 (s, 1H, H-8, Ar); 6.46 (dd, 1H, *J* = *8.6, 2.3* Hz, H-6, Ar); 6.94 (d, 1H, *J* = 8.6 Hz, H-5, Ar); 8.56 (s, CH=N). ^13^C NMR (DMSO-*d_6_*) δ: 12.8 (2-CH_3_); 14.3 (CH_3_); 25.0 (C-4, CH_2_); 44.4 (2-CH_2_); 64.1 (C-3, C-CN); 75.3 (O-CH_2_); 99.4 (C-8, Ar); 103.8 (C-6, Ar); 109.6 (C-4a); 119.3 (CN); 129.8 (C-5, Ar); 148.1 (C-8a); 150.7 (C-7, Ar); 158.7 (C-2, -C=N-); 161.4(C-1′, -O-C=N).

*Ethyl N-(3-cyano-4H-chromen-2-yl)formimidate* (**6b**). According to the standard procedure, **6b** was synthesized from 2-amino-4*H*-1-chromene-3-carbonitrile **4b** (2.5 mmol) in 69% yield as yellow powder; mp = 142–144 °C. IR (KBr, ν, cm^−1^): 1510 (C=C); 2209 (CN); 1646 (C=N). ^1^H NMR (DMSO-*d_6_*) δ: 1.33 (t, 3H, *J* = 7.1 Hz, CH_3_); 3.86 (s, 2H, CH_2_); 4.33 (q, 2H, *J* = 7.1 Hz, O-CH_2_); 3.50 (s, 1H, H-4); 4.33 (q, 2H, *J* = 7.1 Hz, CH_2_); 7.08 (d, 1H, *J* = 8.1 Hz, H-8, Ar); 7.14 (t, 1H, *J* = 7.3 Hz, H-6, Ar); 7.21 (d, 1H, *J* = 7.2Hz, H-5, Ar); 7.26 (t, 1H, *J* = 7.3 Hz, H-7, Ar); 8.56 (s, CH=N). ^13^C NMR (DMSO-*d_6_*) δ:13.9 (CH_3_); 28.3 (C-4, CH_2_); 64.0 (O-CH_2_); 74.7 (C-3, C-CN); 117.0 (C-8, Ar); 118.4 (C-4a); 119.0 (CN); 125.5 (C-6, Ar); 128.7 (C-5, Ar); 129.3 (C-7, Ar); 149.7 (C-8a); 159.0 (C-2, -C=N-); 161.3 (C-1′, -O-C=N).

*Ethyl N-(3-cyano-8-methoxy-4H-chromen-2-yl)formimidate* (**6c**). According to the standard procedure, **6c** was synthesized from 2-amino-8-methoxy-4*H*-1-chromene-3-carbonitrile **4c** (2.5 mmol) in 52% yield as yellow powder; mp = 153–155 °C. IR (KBr, ν, cm^−1^): 1585 (C=C); 2212 (CN); 1673 (C=N). ^1^H NMR (DMSO-*d_6_*) δ: 1.32 (m, 3H, CH_3_); 3.64 (s, 2H, CH_2_); 3.81(s, 3H, O-CH_3_); 4.33 (dd, 2H, *J* = 12.0, 5.1 Hz, O-CH_2_); 6.73 (d, 1H, *J* = 7.1 Hz, H-7, Ar); 6.95 (d, 1H, *J* = 7.6Hz, H-5, Ar); 7.06 (m, 1H, H-6, Ar); 8.40 (s, CH=N). ^13^C NMR (DMSO-*d_6_*) δ: 14.6 (CH_3_); 26.3 (C-4, CH_2_); 56.7 (O-CH_3_); 64.7 (O-CH_2_); 74.7 (C-3, C-CN); 111.6 (C-7, Ar); 118.9 (CN); 119.2 (C-5, Ar); 120.3 (C-4a); 125.3 (C-6, Ar); 139.0 (C-8a); 148.0 (C-8, Ar); 158.3 (C-2, -C=N-); 161.3 (C-1′, -O-C=N).

*Ethyl N-(2-cyano-1H-naphto[2,1-b]pyran-3-yl)formimidate* (**6d**). According to the standard procedure, **6d** was synthesized from 3-amino-1*H*-naphto[2,1*-b*]pyran-2-carbonitrile **4d** (2.5 mmol) in 90% yield as yellow powder; mp = 124–126 °C. IR (KBr, ν, cm^−1^): 1560 (C=C); 2212 (CN); 1674 (C=N). ^1^H NMR (DMSO-*d_6_*) δ: 1.33 (t, 3H, *J* = 6.7 Hz, CH_3_); 3.91 (s, 2H, CH_2_); 4.33 (dd, 2H, *J* = 12.1, 6.0 Hz, O-CH_2_); 7.26 (d, 1H, *J* = 8.7 Hz, H-8, Ar); 7.52 (t, 1H, *J* = 7.0 Hz, H-5”, Ar); 7.61 (t, 1H, *J* = 7.0 Hz, H-6”, Ar); 7.79 (d, 1H, *J* = 7.9 Hz, H-6, Ar); 7.86 (d, 1H, *J* = 8.7 Hz, H-7, Ar); 7.92 (d, 1H, *J* = 7.7 Hz, H-5′, Ar); 8.58 (s, CH=N). ^13^C NMR (DMSO-*d_6_*) δ: 14.1 (CH_3_); 23.3 (C-4, CH_2_); 64.6 (O-CH_2_); 75.1 (C-3, C-CN); 111.2 (C-8, Ar); 117.9 (C-4a); 119.5 (CN); 123.6 (C-5”, Ar); 126.1 (C-6”, Ar); 127.7 (C-7, Ar); 128.7 (C-6′, Ar); 129.4 (C-5′, Ar); 131.0 (C-5, Ar); 131.1 (C-6, Ar); 146.8 (C-8a); 157.9 (C-2, -C=N-); 162.0 (C-1′, -O-C=N).

#### 4.1.2. Standard Procedure for Synthesis of 3-Amino-4-imino-3,5-dihydro-4*H*-chromeno[2,3-*d*]pyrimidine **8(a-c)** and 10-Amino-11-imino-11*H*-benzo[5,6]chromeno[2,3-*d*]pyrimidine **8d**

In a 50 mL round-bottomed flask provided with a magnetic stirrer and condenser, a mixture of ethyl *N*-(3-cyano-4*H*-chromen-2-yl)formimidate **6** (10 mmol) and hydrazine hydrate 61% *w*/*v* (0.8 g, 0.78 mL, 16 mmol) in 40 mL of absolute ethanol was stirred over a period of 2 h at room temperature (monitored by thin layer chromatography on silica plates 60 F254 Merck with appropriate eluent). As the reaction progresses, the formation of an insoluble material in the reaction medium is observed. The desired insoluble compound **8** was collected using filtration on a Büchner funnel (porosity N° 4). This collected solid material was washed carefully (2 × 5 mL) with cooled absolute ethanol (initially stocked at 4 °C) and then it was dried under high vacuum (10^−2^ Torr) at 45 °C for 60 min, which afforded the desired compound **8**. It can be used further without purification after control of the quality of the product by ^1^H NMR in DMSO-*d_6_*. If necessary, purification by recrystallization can be conducted from a minimum of absolute ethanol, which gives pure compound **8**.

*3-Amino-7-diethylamino-4-imino-3,5-dihydro-4H-chromeno[2,3-d]pyrimidine* (**8a**). According to the standard procedure, **8a** was synthesized from ethyl *N*-(3-cyano-7-diethylamino-4*H*-chromen-2-yl)formimidate **6a** (10 mmol) in 50% yield as a yellowish powder; mp = 212–214 °C. IR (KBr, ν, cm^−1^): 1560 (C=C); 1644 (C=N); 3260 (NH); 3278–3320 (NH_2_). ^1^H NMR (DMSO-*d_6_*) δ: 1.07 (t, 6H, *J* = 6.9 Hz, 2-CH_3_); 3.31 (dd, 4H, *J* = 14, 7.1 Hz, 2-CH_2_); 3.49 (s, 2H, CH_2_); 5.69 (br s, NH_2_); 6.27 (s, 1H, H-9, Ar); 6.45 (dd, 1H, *J* = 8.5, 2.3 Hz, H-6, Ar); 7.01 (s, 1H, H-7, Ar); 8.01 (s, CH=N). ^13^C NMR (DMSO-*d_6_*) δ: 12.2 (2-CH_3_); 22.7 (C-5, CH_2_); 44.2 (2-CH_2_); 95.5 (C-4a); 99.2 (C-9, Ar); 105.1 (C-5a); 108.9 (C-7, Ar); 130.1 (C-6, Ar); 147.9 (C-2, -C=N); 150.0 (C=NH); 151.1 (C-9a); 156.2 (C-8, C-N, Ar).

*3-Amino-4-imino-3,5-dihydro-4H-chromeno[2,3-d]pyrimidine* (**8b**). According to the standard procedure, **8b** was synthesized from ethyl *N*-(3-cyano-4*H*-chromen-2-yl)formimidate **6b** (10 mmol) in 69% yield as yellowish powder; mp = 240–242 °C. IR (KBr, ν, cm^−1^): 1581 (C=C); 1646 (C=N); 3160 (NH); 3330–3445 (NH_2_). ^1^H NMR (DMSO-*d_6_*) δ: 3.62 (s, 2H, CH_2_); 5.68 (br s, NH_2_); 7.21 (m, 4H, H-6, H-7, H-8, H-9, Ar); 7.99 (s, CH=N). ^13^C NMR (DMSO-*d_6_*) δ: 23.9 (C-5, CH_2_); 95.1 (C-4a); 116.8 (C-9, Ar); 119.5 (C-5a); 124.9 (C-7, Ar); 128.3 (C-8, Ar); 129.9 (C-6, Ar); 150.3 (C-2, -C=N); 150.5 (C=NH); 156.4 (C-9a).

*3-Amino-4-imino-9-methoxy-3,5-dihydro-4H-chromeno[2,3-d]pyrimidine* (**8c**). According to the standard procedure, **8c** was synthesized from ethyl *N*-(3-cyano-8-methoxy-4*H*-chromen-2-yl)formimidate **6c** (10 mmol) in 76% yield as yellowish powder; mp ≥ 260 °C. IR (KBr, ν, cm^−1^): 1510 (C=C); 1648 (C=N); 3212 (NH); 3296–3331 (NH_2_). ^1^H NMR (DMSO-*d_6_*) δ: 3.36 (s, 2H, CH_2_); 3.81 (s, 3H, OCH_3_); 5.70 (br s, NH_2_); 6.79 (d, 1H, *J* = 7.5 Hz, H-8, Ar); 6.91 (d, 1H, *J* = 8.0 Hz, H-6, Ar); 7.02 (t, 1H, *J* = 7.9 Hz, H-7, Ar); 8.01 (s, CH=N). ^13^C NMR (DMSO-*d_6_*) δ: 23.7 (C-5, CH_2_); 56.2 (CH_3_); 94.9 (C-4a); 111.2 (C-8, Ar); 120.4 (C-5a); 121.1 (C-6, Ar); 124.4 (C-7, Ar); 139.7 (C-9a); 148.0 (C-2, -C=N); 150.3 (C=NH); 156.0 (C-9, Ar).

*10-Amino-11-imino-11H-benzo[5,6]chromeno[2,3-d]pyrimidine* (**8d**). According to the standard procedure, **8d** was synthesized from ethyl *N*-(2-cyano-1*H*-naphto[2,1*-b*]pyran-3-yl)formimidate **6d** (10 mmol) in 90% yield as yellowish powder; mp ≥ 260 °C. IR (KBr, ν, cm^−1^): 1520 (C=C); 1650 (C=N); 3178 (NH); 3280–3301 (NH_2_). ^1^H NMR (DMSO-*d_6_*) δ_:_ 3.91 (s, 2H, CH_2_); 5.76 (br s, NH_2_); 7.28 (d, 1H, *J* = 8.8 Hz, H-6, Ar); 7.52 (t, 1H, *J* = 7.1 Hz, H-3, Ar); 7.64 (t, 1H, *J* = 7.4 Hz, H-2, Ar); 7.86 (d, 1H, *J* = 8.8 Hz, H-5, Ar); 7.94 (d, 1H, *J* = 7.9 Hz, H-4, Ar); 8.00 (d, 1H, *J* = 8.2 Hz, H-1, Ar); 8.09 (s, CH=N). ^13^C NMR (DMSO-*d_6_*) δ: 21.8 (C-12, CH_2_); 95.0 (C-11a); 111.9 (C-6, Ar); 117.6 (C-12a); 123.6 (C-5, Ar); 125.4 (C-1, Ar); 127.6 (C-2, Ar); 128.7 (C-3, Ar); 129.1 (C-4, Ar); 130.7 (C-12b, Ar); 131.9 (C-4a, Ar); 147.4 (C-9, -C=N); 150.2 (C=NH); 155.8 (C-6a).

#### 4.1.3. Standard Procedure for Microwave Assisted Organic Synthesis of 3-(*N*-Phenylmethylidene)amino-4-imino-3,5-dihydro-4*H*-chromeno[2,3-*d*]pyrimidine **10(a-i)** and 10-*N*-(Phenylmethylidene)amino-11-imino-11*H*-naphto[1′,2′:5,6]pyrano[2,3-*d*]pyrimidine **10(j-l)** in the Monowave 300^®^ Microwave Reactor

In a cylindrical quartz reactor (Ø = 1.8 cm, 20 mL) 3-amino-4-imino-3,5-dihydro-4*H*-chromeno[2,3*-d*]pyrimidine **8** (2.05 mmol) or 10-amino-11-imino-11*H*-benzo[5,6]chromeno[2,3*-d*]pyrimidine **8d** (2.05 mmol), commercial aromatic aldehyde **9** (2.46 mmol, 1.2 equiv.), piperidine (0.5% mol, 1.1 mg, 1.3 μL, 12.5 μmol), absolute ethanol (12 mL) and a magnetic stirring bar were introduced successively. The glass tube was sealed with a snap cap and placed in the Monowave 300^®^ microwave cavity (P = 800 Watt). The stirred mixture was irradiated at 160 °C (after a ramp of 4 min from 20 to 160 °C) over a period of 60 min. After microwave dielectric heating, the crude reaction mixture was allowed to cool down at room temperature for 30–45 min or more until the complete formation of insoluble product in the crude reaction mixture. The resulting insoluble product **10** was collected using filtration on a Büchner funnel (porosity N° 4), washed carefully with diethyl ether Et_2_O (2 × 10 mL), then dried under high vacuum (10^−2^ Torr) at room temperature for 60 min which resulted in the desired compound **10** as powder. Recrystallization from a minimum of absolute ethanol gave pure compound **10**.

*(3E) 3-N-(4-Dimethylaminophenylmethylidene)amino-7-diethylamino-4-imino-3,5-dihydro-4H-chromeno[2,3-d]pyrimidine* (**10a**). According to the standard procedure, **10a** was synthesized, respectively, from 3-amino-7-diethylamino-4-imino-3,5-dihydro-4*H*-chromeno[2,3*-d*]pyrimidine **8a** (2.05 mmol) and commercial 4-dimethylaminobenzaldehyde **9a** (367 mg, 2.46 mmol, 1.2 equiv.) in 50% yield as yellowish powder; mp = 240–242 °C. IR (KBr, ν, cm^−1^): 1560 (C=C); 1623 (C=N); 3186 (NH). ^1^H NMR (DMSO-*d_6_*) δ: 1.09 (s, 6H, 2-CH_3_); 2.97 (s, 6H, NMe_2_); 3.35 (s, 4H, 2-CH_2_); 5.95 (s, 2H, CH_2_); 6.47 (d, 2H, *J* = 6.8 Hz, H-7, Ar); 6.77 (d, 2H, *J* = 7.1 Hz, H-3′, H-5′, Ar); 7.04 (d, 1H, *J* = 7.7 Hz, H-6, Ar); 7.52 (d, 2H, *J* = 7.1 Hz, H-2′, H-6′, Ar); 8.23 (d, 1H, CH=N); 10.52 (br s, 1H, NH). ^13^C NMR (DMSO-*d_6_*) δ: 12.9 (2-CH_3_); 22.9 (C-5, CH_2_); 40.3 (2-CH_3_); 44.2 (2-CH_2_); 93.6 (C-4a); 99.2 (C-9, Ar); 105.6 (C-7, Ar); 108.9 (C-5a); 112.4 (C-3′, C-5′, Ar); 122.7 (C-1′, Ar); 128.4 (C-2′, C-6′, Ar); 130.1 (C-6, Ar); 145.4 (C=NH); 148.0 (N-CH=N); 151.1 (C-9a); 151.6 (C-8, Ar); 156.5 (C-10a); 159.1 (C-4′, Ar); 163.6 (N-C=N). HRMS (ES^+^, MeOH/CH_2_Cl_2_ 9:1), *m/z:* 346.1663 found (calculated for C_20_H_20_N_5_O, [M+H]^+^ requires 346.16624).

*(3E) 3-N-(2-Hydroxyphenylmethylidene)amino-7-diethylamino-4-imino-3,5-dihydro-4H-chromeno[2,3-d]pyrimidine* (**10b**). According to the standard procedure, **10b** was synthesized, respectively, from 3-amino-7-diethylamino-4-imino-3,5-dihydro-4*H*-chromeno[2,3*-d*]pyrimidine **8a** (2.05 mmol) and commercial 2-hydroxybenzaldehyde **9c** (or salicylaldehyde **1b**) (300 mg, 2.46 mmol, 1.2 equiv.) in 65% yield as yellowish powder; mp = 236–238 °C. IR (KBr, ν, cm^−1^): 1598 (C=C); 1637 (C=N); 3170 (NH); 3310 (OH). ^1^H NMR (DMSO-*d_6_*) δ: 1.07 (t, 6H, *J* = 6.9 Hz, 2-CH_3_); 3.29 (q, 4H, *J* = 13.5, 6.6 Hz, 2-CH_2_); 3.78 (s, 2H, CH_2_); 6.31 (d, 1H, *J* = 1.9 Hz, H-9, Ar); 6.44 (dd, 1H, *J* = 8.5, 2 Hz, H-7, Ar); 6.92 (d, 2H, *J* = 7.7 Hz, H-2′, H-4′, Ar); 7.00 (d, 1H, *J* = 8.6 Hz, H-5′, Ar); 7.25 (t, 1H, *J* = 7.7 Hz; H-3′, Ar); 7.47 (d, 2H, *J* = 7.4 Hz, H-6, Ar); 8.31 (s, 1H, CH=N), 8.52 (s, 1H, N-CH=N); 10.89 (br s, 1H, NH); 11.55 (br s, 1H, OH). ^13^C NMR (DMSO-*d_6_*) δ: 13.2 (2-CH_3_); 22.1 (C-5, CH_2_); 44.2 (2-CH_2_); 94.1 (C-4a); 99.4 (C-9, Ar); 105.1 (C-7, Ar); 109.2 (C-5a); 117.0 (C-5′, Ar); 119.4 (C-1′, Ar); 119.6 (C3′, Ar); 129.8 (C-6′, Ar); 130.4 (C-4′, Ar); 130.9 (C-6, Ar); 145.5 (C=NH); 148.0 (N-CH=N); 151.0 (C-9a); 156.7 (C-8, Ar); 157.7 (C=N); 158.3 (C-2′, Ar); 163.3 (C-10a). HRMS (ES^+^, MeOH/CH_2_Cl_2_ 9:1), *m/z:* 390.1922 found (calculated for C_22_H_24_N_5_O_2_, [M+H]^+^ requires 390.19245).

*(3E) 3-N-(Phenylmethylidene)amino-7-diethylamino-4-imino-3,5-dihydro-4H-chromeno[2,3-d]pyrimidine* (**10c**). According to the standard procedure, **10c** was synthesized, respectively, from 3-amino-7-diethylamino-4-imino-3,5-dihydro-4*H*-chromeno[2,3*-d*]pyrimidine **8a** (2.05 mmol) and commercial benzaldehyde **9b** (261 mg, 2.46 mmol, 1.2 equiv.) in 60% yield as yellowish powder; mp = 250–252 °C. IR (KBr, ν, cm^−1^): 1558 (C=C); 1639 (C=N); 3197 (NH). ^1^H NMR (DMSO-*d_6_*) δ: 1.08 (t, 6H, *J* = 6.9 Hz, 2-CH_3_); 3.34 (q, 4H, *J* = 13.7, 6.8 Hz, 2-CH_2_); 3.98 (s, 2H, CH_2_); 6.33 (s, 1H, H-9, Ar); 6.47 (dd, 1H, *J =* 10.2 Hz, H-7, Ar); 7.05 (d, 1H, *J =* 8.5 Hz, H-6, Ar); 7.40 (m, 1H, H-4′, Ar); 7.46 (t, 2H, *J =* 7.3 Hz, H-3′, H-5′, Ar); 7.70 (d, 2H, *J =* 7.3 Hz, H-2′, H-6′, Ar); 8.28 (d, 2H, CH=N, N-CH=N); 10.87 (br s, 1H, NH). ^13^C NMR (DMSO-*d_6_*) δ: 13.8 (2-CH_3_); 22.6 (C-5, CH_2_); 44.4 (2-CH_2_); 94.4 (C-4a); 99.2 (C-9, Ar); 105.4 (C-7, Ar); 108.9 (C-5a); 127.3 (C-2′, C-6′, Ar); 129.3 (C-3′, C-5′, Ar); 129.9 (C-4′, Ar); 130.2 (C-6, Ar); 135.1 (C-1′, Ar); 144.5 (C=NH); 148.0 (N-CH=N); 151.0 (C-9a); 156.4 (C-8, C-N, Ar); 159.1 (C=N); 163.7 (C-10a). HRMS (ES^+^, MeOH/CH_2_Cl_2_ 9:1), *m/z*: 374.1977 found (calculated for C_22_H_24_N_5_O, [M+H]^+^ requires 374.1975).

*(3E) 3-N-(4-Dimethylaminophenylmethylidene)amino-4-imino-3,5-dihydro-4H-chromeno[2,3-d]pyrimidine* (**10d**). According to the standard procedure, **10d** was synthesized, respectively, from 3-amino-4-imino-3,5-dihydro-4*H*-chromeno[2,3*-d*]pyrimidine **8b** (2.05 mmol) and commercial 4-dimethylaminobenzaldehyde **9a** (367 mg, 2.46 mmol, 1.2 equiv.) in 43% yield as a yellowish powder; mp ≥ 260 °C. IR (KBr, ν, cm^−1^): 1540 (C=C); 1630 (C=N); 3489 (NH). ^1^H NMR (DMSO-*d_6_*) δ: 2.97 (s, 6H, 2-CH_3_); 4.13 (s, 2H, CH_2_); 6.76 (d, 2H, *J* = 8.7 Hz, H-3′, H-5′, Ar); 7.11 (m, 2H, H-7, H-9, Ar); 7.27 (m, 2H, H-6, H-8, Ar); 7.52 (d, 2H, *J* = 7.1 Hz, H-2′, H-6′, Ar); 8.15 & 8.24 (s, 2H, N-CH=N, CH=N); 10.64 (br s, 1H, NH). ^13^C NMR (DMSO-*d_6_*) δ: 24.1 (C-5, CH_2_); 39.3 (2-CH_3_); 93.3 (C-4a); 112.4 (C-3′, C-5′, Ar); 116.9 (C-9, Ar); 120.1 (C-1′, Ar); 122.6 (C-5a); 124.7 (C-7, Ar); 128.4 (C-2′, C-6′, Ar); 128.6 (C-8, Ar); 128.8 (C-6, Ar); 145.5 (C=N); 150.2 (N-C=N); 151.6 (C-9a); 156.6 (C-4′, Ar); 159.0 (C=N); 163.5 (C-10a). HRMS (ES^+^, MeOH/CH_2_Cl_2_ 9:1), *m/z*: 346.1662 found (calculated for C_20_H_19_N_5_O, [M+H]^+^ requires 346.16624).

*(3E) 3-N-(2-Hydroxyphenylmethylidene)amino-4-imino-3,5-dihydro-4H-chromeno[2,3-d]pyrimidine* (**10e**). According to the standard procedure, **10e** was synthesized, respectively, from 3-amino-4-imino-3,5-dihydro-4*H*-chromeno[2,3*-d*]pyrimidine **8b** (2.05 mmol) and commercial 2-hydroxybenzaldehyde **9c** (or salicylaldehyde **1b**) (300 mg, 2.46 mmol, 1.2 equiv.) in 65% yield as a yellowish powder; mp ≥ 260 °C. IR (KBr, ν, cm^−1^): 1520 (C=C); 1630 (C=N); 3189 (NH); 3246 (OH). ^1^H NMR (DMSO-*d_6_*) δ: 4.01 (s, 2H, CH_2_); 6.93 (dd, 2H, *J* = 7.6, 5.0 Hz, H-6, H-9, Ar); 7.16 (dd, 2H, *J* = 13.6, 7.4 Hz, H-7, H-8, Ar); 7.28 (t, 3H, *J* = 7.6 Hz, H-3′, H-4′, H-5′, Ar); 7.52 (d, 1H, *J* = 7.6 Hz, H-2′, Ar); 8.34 & 8.56 (s, 2H, N-CH=N, CH=N); 11.02 (br s, 1H, OH); 11.45 (br s, 1H, NH). ^13^C NMR (DMSO-*d_6_*) δ: 22.9 (C-5, CH_2_); 93.7 (C-4a); 116.9 (C-9, Ar); 117.1 (C-5′, Ar); 119.5 (C-1′, Ar); 119.6 (C-3′, Ar); 119.8 (C-5a); 124.9 (C-7′, Ar); 129.7 (C-2′, Ar); 129.8 (C-6, Ar); 131.2 (C-4′, Ar); 145.5 (C=NH); 150.3 (N-C=N); 157.0 (C-9a); 157.6 (C=N); 158.4 (C-6′, Ar); 163.1 (C-10a). HRMS (ES^+^, MeOH/CH_2_Cl_2_ 9:1), *m/z*: 319.1189 found (calculated for C_18_H_15_N_4_O_2_, [M+H]^+^ requires 319.11895).

*(3E) 3-N-(Phenylmethylidene)amino-4-imino-3,5-dihydro-4H-chromeno[2,3-d]pyrimidine* (**10f**). According to the standard procedure, **10f** was synthesized, respectively, from 3-amino-4-imino-3,5-dihydro-4*H*-chromeno[2,3*-d*]pyrimidine **8b** (2.05 mmol) and commercial benzaldehyde **9b** (261 mg, 2.46 mmol, 1.2 equiv.) in 30% yield as yellowish powder; mp = 232–234 °C. IR (KBr, ν, cm^−1^): 1520 (C=C); 1638 (C=N); 3199 (NH). ^1^H NMR (DMSO-*d_6_*) δ: 4.01 (s, 2H, CH_2_); 6.93 (dd, 2H, *J* = 7.1, 4.4 Hz, H-7, H-8); 7.14 (d, 1H, *J* = 8.1 Hz, H-9); 7.17 (d, 1H, *J* = 8.2 Hz, H-4′); 7.28 (m, 4H, H-2′, H-3′, H-5′, H-6′, Ar); 7.52 (d, 1H, *J* = 7.8 Hz, H-6, Ar); 8.34 & 8.56 (s, 2H, N-CH=N, CH=N); 11.02 (br s, 1H, NH). ^13^C NMR (DMSO-*d_6_*) δ: 22.8 (C-5, CH_2_); 93.6 (C-4a); 116.9 (C-9, Ar); 117.1 (C-5a); 119.6 (C-7, Ar); 119.7 (C-2′, C-6′, Ar); 124.9 (C-3′, C-5′, Ar); 128.7 (C-8, Ar); 129.7 (C-6, Ar); 129.8 (C-4′, Ar); 131.2 (C-1′, Ar); 150.3 (C=NH); 157.0 (N-C=N); 157.7 (C-9a); 158.4 (C-10a); 163.2 (C=N). HRMS (ES^+^, MeOH/CH_2_Cl_2_ 9:1), *m/z*: 303.1239 found (calculated for C_18_H_15_N_4_O, [M+H]^+^ requires 303.12404).

*(3E) 3-N-(4-Dimethylaminophenylmethylidene)amino-4-imino-9-methoxy-3,5-dihydro-4H-chromeno[2,3-d]pyrimidine* (**10g**). According to the standard procedure, **10g** was synthesized, respectively, from 3-amino-4-imino-9-methoxy-3,5-dihydro-4*H*-chromeno[2,3*-d*]pyrimidine **8c** (2.05 mmol) and commercial 4-dimethylaminobenzaldehyde **9a** (367 mg, 2.46 mmol, 1.2 equiv.) in 60% yield as a yellowish powder; mp ≥ 260 °C. IR (KBr, ν, cm^−1^): 1550 (C=C); 1641 (C=N); 3199 (NH). ^1^H NMR (DMSO-*d_6_*) δ: 2.97 (s, 6H, 2-CH_3_); 3.84 (s,3H, OCH_3_); 4.12 (s, 2H, CH_2_); 6.76 (d, 2H, *J* = 8.7 Hz, H-3′, H-5′, Ar); 6.82 (d, 1H, *J* = 7.6 Hz, H-8, Ar); 6.95 (d, 1H, *J* = 8.0 Hz, H-6, Ar); 7.06 (t, 1H, *J =* 7.9 Hz, H-7, Ar); 7.51 (d, 2H, *J* = 8.3 Hz, H-2′, H-6′, Ar); 8.14 & 8.22 (s, 1H, N-CH=N, CH=N); 10.62 (br s, 1H, NH). ^13^C NMR (DMSO-*d_6_*) δ: 24.0 (C-5, CH_2_); 56.4 (2-CH_3_); 93.1 (CH_3_); 111.3 (C-4a); 112.4 (C-3′, C-5′, Ar); 120.9 (C-2′, C-6′, Ar); 122.6 (C-1′, Ar); 122.9 (C-6, Ar); 124.4 (C-5a); 128.7 (C-7, Ar); 139.6 (C-8, Ar); 145.1 (C=NH); 148.1 (C-9a); 151.8 (N-C=N); 156.5 (C-9, Ar); 159.0 (C-4′, Ar); 163.6 (C=N). HRMS (ES^+^, MeOH/CH_2_Cl_2_ 9:1), *m/z*: 376.1767 found (calculated for C_21_H_22_N_5_O_2_, [M+H]^+^ requires 376.1768).

*(3E) 3-N-(2-Hydroxyphenylmethylidene)amino-4-imino-9-methoxy-3,5-dihydro-4H-chromeno[2,3-d]pyrimidine* (**10h**). According to the standard procedure, **10h** was synthesized, respectively, from 3-amino-4-imino-9-methoxy-3,5-dihydro-4*H*-chromeno[2,3*-d*]pyrimidine **8c** (2.05 mmol) and commercial 2-hydroxybenzaldehyde **9c** (or salicylaldehyde **1b**) (300 mg, 2.46 mmol, 1.2 equiv.) in 90% yield as yellowish powder; mp ≥ 260 °C. IR (KBr, ν, cm^−1^): 1590 (C=C); 1650 (C=N); 3190 (NH); 3370 (OH). ^1^H NMR (DMSO-*d_6_*) δ: 3.83 (s, 3H, OCH_3_); 3.94 (s, 2H, CH_2_); 6.79 (d, 1H, *J* = 7.5 Hz, H-8, Ar); 6.92 (m, 3H, H-2′, H-3′, H-5′, Ar); 7.05 (t, 1H, *J* = 7.9 Hz, H-4′, Ar); 7.26 (t, 1H, *J* = 7.7 Hz, H-7, Ar); 7.48 (d, 2H, *J* = 7.0 Hz, H-6, Ar); 8.31 & 8.52 (s, 2H, N-CH=N, CH=N); 10.96 (br s, 1H, OH); 11.46 (br s, 1H, NH). ^13^C NMR (DMSO-*d_6_*) δ: 22.1 (C-5, CH_2_); 56.3 (CH_3_); 93.43 (C-4a); 111.6 (C-8, Ar); 116.8 (C-5′, Ar); 119.5 (C-1′, Ar); 119.7 (C-3′, Ar); 120.2 (C-6, Ar); 120.8 (C-5a); 124.5 (C-7, Ar); 129.7 (C-2′, Ar); 131.1 (C-4′, Ar); 139.6 (C=NH); 145.5 (C-9a); 148.2 (N-C=N); 156.8 (C-9, Ar); 157.7 (C=N); 158.3 (C-6′; Ar); 163.0 (C-10a). HRMS (ES^+^, MeOH/CH_2_Cl_2_ 9:1), *m/z*: 349.1295 found (calculated for C_19_H_17_N_4_O_3_, [M+H]^+^ requires 349.12952).

*(3E) 3-N-(Phenylmethylidene)amino-4-imino-9-methoxy-3,5-dihydro-4H-chromeno[2,3-d]pyrimidine* (**10i**). According to the standard procedure, **10i** was synthesized, respectively, from 3-amino-4-imino-9-methoxy-3,5-dihydro-4*H*-chromeno[2,3*-d*]pyrimidine **8c** (2.05 mmol) and commercial benzaldehyde **9b** (261 mg, 2.46 mmol, 1.2 equiv.) in 51% yield as yellowish powder; mp = 252–254 °C. IR (KBr, ν, cm^−1^): 1550 (C=C); 1645 (C=N); 3196 (NH). ^1^H NMR (DMSO-*d_6_*) δ: 3.84 (s,3H, OCH_3_); 4.15 (s, 2H, CH_2_); 6.84 (d, 1H, *J* = 7.5 Hz, H-8, Ar); 6.96 (d, 1H, *J* = 7.9 Hz, H-6, Ar); 7.08 (t, 1H, *J* = 7.9 Hz, H-7, Ar); 7.44 (m, 3H, H-3′, H-4′, H-5′, Ar); 7.71 (d, 2H, *J* = 7.3 Hz, H-2′, H-6′ Ar); 8.28 (s, 2H, N-CH=N, CH=N); 10.96 (br s, 1H, NH). ^13^C NMR (DMSO-*d_6_*) δ: 24.1 (C-5, CH_2_); 56.2 (CH_3_); 93.8 (C-4a); 120.7 (C-5a); 120.9 (C-8, Ar); 124.5 (C-6, Ar); 126.9 (C-2′, C-6′, Ar); 129.1 (C-3′, C-5′, Ar); 129.9 (C-7, Ar); 135.2 (C-4′, Ar); 139.5 (N-C=N); 144.4 (C-1′, Ar); 148.0 (C=N); 156.5 (C=NH); 159.0 (C-9a); 163.5 (C-9, Ar). HRMS (ES^+^, MeOH/CH_2_Cl_2_ 9:1), *m/z*: 333.1349 found (calculated for C_19_H_17_N_4_O_2_, [M+H]^+^ requires 333.1346).

*(10E) 10-N-(4-Dimethylaminophenylmethylidene)amino-11-imino-11H-naphto[1′,2′:5,6]pyrano[2,3-d]pyrimidine* (**10j**). According to the standard procedure, **10j** was synthesized, respectively, from 10-amino-11-imino-11*H*-benzo[5,6]chromeno[2,3*-d*]pyrimidine **8d** (2.05 mmol) and commercial 4-dimethylaminobenzaldehyde **9a** (367 mg, 2.46 mmol, 1.2 equiv.) in 60% yield as yellowish powder; mp ≥ 260 °C. IR (KBr, ν, cm^−1^): 1510 (C=C); 1630 (C=N); 3191 (NH). ^1^H NMR (DMSO-*d_6_*) δ: 3.00 (s, 6H, 2-CH_3_); 4.34 (s, 2H, CH_2_); 6.81 (d, 2H, *J* = 8.8 Hz, H-4′, H-5′, Ar); 7.36 (d, 1H, *J* = 8.9 Hz, H-6, Ar); 7.56 (m, 1H, H-4 Ar); 7.60 (d, 2H, *J* = 8.8 Hz, H-2′, H-6′ Ar); 7.74 (t, 1H, *J* = 7.6 Hz, H-1, Ar); 7.92 (d, 1H, *J* = 8.9 Hz, H-5, Ar); 8.00 (t, 2H, *J* = 8.3 Hz, H-2, H-3, Ar); 8.25 (s, 1H, CH=N); 8.31 (s, 1H, N-CH=N); 10.66 (br s, 1H, NH). ^13^C NMR (DMSO-*d_6_*) δ: 21.8 (C-12, CH_2_); 31.3 (2-CH_3_); 92.8 (C-11a); 112.5 (C-3′, C-5′, Ar); 118.1 (C-6, Ar); 122.6 (C-12a); 123.2 (C-1′, Ar); 125.4 (C-5, Ar); 127.7 (C-1, Ar); 128.5 (C-2′, C-6′, Ar); 129.0 (C-2, Ar); 129.3 (C-3, Ar); 130.7 (C-4, Ar); 131.8 (C-12b); 145.8 (C-4a); 147.5 (C=NH); 151.8 (N-C=N); 157.0 (C-6a); 159.1 (C-4, Ar); 162.8 (C=N). HRMS (ES^+^, MeOH/CH_2_Cl_2_ 9:1), *m/z*: 396.1822 found (calculated for C_24_H_22_N_5_O, [M+H]^+^ requires 396.1819).

*(10E) 10-N-(2-Hydroxyphenylmethylidene)amino-11-imino-11H-naphto[1′,2′:5,6]pyrano[2,3-d]pyrimidine* (**10k**). According to the standard procedure, **10k** was synthesized, respectively, from 10-amino-11-imino-11*H*-benzo[5,6]chromeno[2,3*-d*]pyrimidine **8d** (2.05 mmol) and commercial 2-hydroxybenzaldehyde **9c** (or salicylaldehyde **1b**) (300 mg, 2.46 mmol, 1.2 equiv.) in 80% yield as yellowish powder; mp ≥ 260 °C. IR (KBr, ν, cm^−1^): 1583 (C=C); 1634 (C=N); 3191 (NH); 3390 (OH). ^1^H NMR (DMSO-*d_6_*) δ: 4.18 (s, 2H, CH_2_); 6.95 (m, 2H, H-4′, H-6′, Ar); 7.32 (m, 2H, H-3′, H-5′, Ar); 7.52 (m, 2H, H-2, H-3 Ar); 7.69 (t, 1H, *J* = 7.6 Hz, H-5, Ar); 7.88 (d, 1H, *J* = 8.9 Hz, H-6, Ar); 7.96 (dd, 2H, *J* = 3.35, 8.1 Hz, H-1, H-4, Ar); 8.37 (s, 1H, CH=N); 8.59 (s, 1H, N-CH=N); 11.00 (br s, 1H, OH); 11.39 (br s, 1H, NH). ^13^C NMR (DMSO-*d_6_*) δ: 20.9 (C-12, CH_2_); 93.3 (C-11a); 112.1 (C-6, Ar); 117.0 (C-12a); 117.9 (C-3′, Ar); 119.8 (C-1′, Ar); 123.3 (C-5′, Ar); 125.5 (C-5, Ar); 127.6 (C-1, Ar); 128.9 (C-2, C-3, Ar); 129.3 (C-4, Ar); 130.7 (C-6′, Ar); 131.2 (C-4′, Ar); 131.8 (C-12b, Ar); 147.4 (C-4a, Ar); 157.0 (C=NH); 157.6 (N-C=N); 158.5 (C-a); 162.6 (C-2′, Ar). HRMS (ES^+^, MeOH/CH_2_Cl_2_ 9:1), *m/z*: 369.1345 found (calculated for C_22_H_17_N_4_O_2_, [M+H]^+^ requires 369.1346).

*(10E) 10-N-(Phenylmethylidene)amino-11-imino-11H-naphto[1′,2′:5,6]pyrano[2,3-d]pyrimidine* (**10l**). According to the standard procedure, **10l** was synthesized, respectively, from 10-amino-11-imino-11*H*-benzo[5,6]chromeno[2,3*-d*]pyrimidine **8d** (2.05 mmol) and commercial benzaldehyde **9b** (261 mg, 2.46 mmol, 1.2 equiv.) in 55% yield as yellowish powder; mp ≥ 260 °C. IR (KBr, ν, cm^−1^): 1572 (C=C); 1615 (C=N); 3112 (NH). ^1^H NMR (DMSO-*d_6_*) δ: 4.30 (s, 2H, CH_2_); 7.30 (d, 1H, *J* = 8.9 Hz, H-6, Ar); 7.48 (m, 4H, H-2′, H-3′, H-5′, H-6′, Ar); 7.67 (t, 1H, *J* = 8.9 Hz, H-4′, Ar); 7.77 (d, 2H, *J* = 7.4 Hz, H-2, H-3, Ar); 7.86 (d, 1H, *J* = 8.9 Hz, H-5, Ar); 7.94 (d, 2H, *J* = 8.2 Hz, H-1, H-4, Ar); 8.33 (s, 2H, N-CH=N, CH=N); 10.98 (br s, 1H, NH). ^13^C NMR (DMSO-*d_6_*) δ: 21.8 (C-12, CH_2_); 93.9 (C-11a); 112.6 (C-6, Ar); 118.0 (C-12a); 123.1 (C-5, Ar); 125.4 (C-1, Ar); 127.1 (C-2′, C-6′, Ar); 127.6 (C-4, Ar); 128.9 (C-2, Ar); 129.2 (C-3, Ar); 129.3 (C-3′, C-5′, Ar); 130.0 (C-4′, Ar); 130.7 (C-12b); 131.7 (C-4a); 144.5 (C-1′, Ar); 147.3 (C=NH); 156.7 (N-C=N); 159.3 (C-6a); 162.9 (C=N). HRMS (ES^+^, MeOH/CH_2_Cl_2_ 9:1), *m/z*: 353.1399 found (calculated for C_22_H_17_N_4_O, [M+H]^+^ requires 353.13969).

#### 4.1.4. Drug Likeness

Five filters were used to predict drug-likeness [43] by the Molsoft software and SwissADME program (http://www.swissadme.ch/, accessed on 24 February 2023) via the ChemAxon’s Marvin JS structure drawing tool.

### 4.2. Biology Section

#### 4.2.1. Antiproliferative Assays

##### Cell Culture

Skin diploid fibroblastic cells were provided by BIOPREDIC International Company (Rennes, France). Huh-7D12 (Ref ECACC: 01042712), Caco2 (Ref ECACC: 86010202), MDA-MB-231 (Ref ECACC: 92020424), MDA-MB-468, HCT-116 (Ref ECACC: 91091005), PC3 (Ref ECACC: 90112714), and MCF7 cell lines were obtained from the ECACC collection. PANC1 cell lines, a generous gift of LBAI Inserm 1227 (Brest, France). Cells were grown according to ECACC recommendations [44] in DMEM for Huh-D12, MDA-MB-231, MDA-MB-468 and fibroblast, in EMEM for Caco2, in McCoy’s for HCT-116 and RPMI for PC3 at 37 °C and 5% CO_2_. All culture media with 10% of FBS, 1% of penicillin-streptomycin and 2 mM glutamine.

##### Protocol for Antiproliferative Assays

Chemicals were solubilized in DMSO at a concentration of 10 mM (stock solution) and diluted in a culture medium to the desired final concentrations. The dose-effect cytotoxic assays (IC_50_ determination) were performed by increasing concentrations of each chemical (final concentrations: 0.1 μM, 0.3 μM, 0.9 μM, 3 μM, 9 μM and 25 μM). The toxicity test of the chemicals on these cells was as follows: 2 × 10^3^ cells for HCT-116 cells or 4 × 10^3^ for the other cells were seeded in 96-multi-well plates in triplicate and left for 24 h for attachment, spreading and growing. Then, cells were exposed for 48 h to increasing concentrations of chemicals, ranging from 0.1 μM to 25 μM, in a final volume of 120 μL of culture medium. After 48 h of treatment, cells were washed in PBS and fixed in a cooled 90% ethanol/5% acetic acid solution for 20 min. The nuclei were stained with Hoechst 3342 (B2261 Merck Sigma-Aldrich) and counted. Image acquisition and analysis were performed using automated imaging analysis with a Cellomics Arrayscan VTI/HCS Reader (Thermo/Scientific, Waltham, MA, USA). The survival percentages were calculated as the percentage of cell number after chemical treatment over cell number after DMSO treatment. The IC_50_ was graphically determined.

#### 4.2.2. “Store-Operated Calcium Entry” (SOCE) Assays

##### Reagents

Dimethyl Sulfoxyde (DMSO-Ref D2438) and Thapsigargin (Tg-Ref: T9033) were purchased from Sigma-Aldrich. Fura-2 QBT™ Calcium Kit was purchased from Molecular Devices (Ref: R8198).

##### HEK293 Cell Culture

The HEK293 cell line was maintained in DMEM High Glucose with Sodium Pyruvate (110 mg/mL) and antibiotics (1% of penicillin/streptomycin) in a humidified incubator at 37 °C, supplemented with 5% of CO_2_.

##### Cytosolic Ca^+2^ Protocol

For Ca^2+^ measurement experiments, HEK293 cells were seeded overnight into 96 wells of a black clear bottom plate (Corning Ref: 3603) at 50 × 10^5^ cells/well in 100 µL culture medium. The resting medium was aspirated and cells were loaded with Fura-2 acetoxymethyl ester (Fura-2 QBT™) fluorochrome. 80 µL per well and incubated 1 h at 37 °C, 5% CO_2_. The Fura-2 QBT™ was aspirated and replaced by an equal volume of free Ca^2+^ Hepes-buffered solution (in mM: 135 NaCl, 5 KCl, 1 MgCl_2_, 1 EGTA, 10 Hepes, 10 glucose, pH adjusted at 7.45 with NaOH containing the tested compounds or DMSO control and incubated for 5 min at 37 °C until evaluating calcium level on FlexStation 3™ Instrument.

##### Measurement of Intracellular Calcium Levels

Intracellular calcium levels were evaluated by recording changes in the fluorescence of the dual-wavelength fluorescent calcium-sensitive dye Fura-2AM using the plate reader FlexStation 3™ (Molecular Devices, Sunnyvale, CA, USA). Dual excitation wavelength capability permits ratiometric measurements of Fura-2AM peak fluorescence emission at 510 nm after excitations at 340 and 380. Modifications in the 340/380 ratio reflect changes in intracellular-free Ca^2+^ concentrations. The FlexStation 3™ temperature was set at 37 °C during data acquisition. Thapsigargin (2 µM) solution and Hepes-buffered solution with Ca^2+^ containing (in mM): 135 NaCl, 5 KCl, 1 MgCl_2_, 2 CaCl_2_, 10 Hepes, 10 glucose, pH adjusted at 7.45 with NaOH were added from a 96-well reservoir plate during Calcium Mobilization Assay running at 100 and 750 s, respectively. Experimental setup parameters were optimized (pipette heights, volumes and rate of additions) to minimize disturbance of the cells while ensuring rapid mixing. The data were stored for later analysis by using SoftmaxPro (Molecular Devices), Excel v.16.66.1 and Graph Pad Prism v.8 software.

##### Measurement of Store-Operated Calcium (SOC) Entry

ER Ca^2+^ stores were depleted with 2 μM Thapsigargin an inhibitor of ER SERCA pumps under Ca^2+^-free conditions to determine the magnitude of intracellular Ca^2+^ release from ER. The subsequent addition of extracellular Ca^2+^ allowed the measurement of SOCE. The magnitude of SOCE was estimated as the maximal values of the normalized F340/F380 ratio following Ca^2+^ re-addition.

##### Calculation and Data Analysis

Data analysis was performed using Softmax Pro and Graph Pad Prism. (Graph Pad): Ca^2+^ concentration variations are estimated using the ratio of RFU at 340 and 380 nm (F340/F380) and for each measurement F340/F380 (ΔF) ratio values were normalized to the initial basal ratio before TG addition (F0). F340/F380 (ΔF/F0) normalization is the best estimation of calcium concentration changes. The inhibition results are expressed in terms of the percentage of response compared to the maximum response on the control condition. Inhibition is estimated as follows: % Inhibition = (100 − % response).

#### 4.2.3. In Vivo Zebrafish *Danio rerio* Assays

##### Zebrafish Maintenance and Embryo/Larva Exposure

Zebrafish (transgenic line Tg(kdrl:Hsa.HRAS-mCherry)^s896^) were housed in agreement with the European Union regulations concerning the protection of experimental animals (Directive 2010/63/EU) and approved by the ethics committee (CREEA: Comité Rennais d’Éthique en matière d’Expérimentation Animale) under permit number EEA B-35-040. Zebrafish were raised in our facilities (ImPACcell-Biosit, Rennes) using a ZebTEC Active Blue Stand Alone system (Tecniplast, Buguggiate, Italy) in recirculated water kept at 28.5 °C and spawned under standard conditions. Eggs were collected 2 h post-fertilization (hpf) and kept in glass Petri dishes containing water for subsequent examination under a binocular microscope to confirm fertilization. Developing embryos were distributed in 96-well plates (1 embryo per well) containing 200 μL of E3 embryonic medium (5 mM NaCl, 0.17mM KCl, 0.33 mM CaCl_2_, 0.3 mM MgSO_4_) and kept in an incubator at 28.5 ± 0.5 °C (14/10 h light/dark photocycle). Chemical treatments were performed by adding DMSO alone (negative control), or test molecule. These experiments were performed in four technical replicates.

##### In Vivo Zebrafish *Danio rerio* Toxicity Assays

The experiments were performed using a Zeiss inverted fluorescent microscope (Observer Z1). The zebrafish toxicological test (Zebratox) was realized as follows: the 24 h post-fertilization embryos were arrayed by hand with one embryo per well in black-walled 96-well microplates for fluorescence assays (Perkin Elmer, Waltham, MA, USA). The embryos were treated before hatching from 0 to 72 h post-fertilization. The 20 mM DMSO stock chemicals were diluted in fish water and added to the wells at final concentrations from 0.0 2µM to 5 μM. The negative control (0.25% DMSO) was added to each replicate plate. The experiments were realized in triplicate in three independent plates. The embryos were incubated for 72 h at 28 °C and were imaged, after anesthesia by incubation at 4 °C for 30 min. The zebrafish fertilized eggs were provided by the ImPACcell-Biosit Zebrafish facilities (Rennes, France).

## Data Availability

The data presented in this study are available on request from the corresponding author.

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
