# Peer review of "Design and Synthesis of Novel N-Benzylidene Derivatives of 3-Amino-4-imino-3,5-dihydro-4H-chromeno[2,3-d]pyrimidine under Microwave, In Silico ADME Predictions, In Vitro Antitumoral Activities and In Vivo Toxicity"

_pharmaceuticals, 2024, doi:10.3390/ph17040458_

Round 1
Reviewer 1 Report
Comments and Suggestions for Authors
Dear Authors,
Following observations were made while reviewing the submitted Manuscript;
1. The title is too. It lacks the novelty. Words like in vivo toxicity and In-silico ADME study can be removed from the title. Make the title more concise and interesting.
2. Abstract containing too many Keywords. Few keywords can be removed (such as cytotoxicity and in silico ADME predictions)
3. HRMS Data of intermediates are missing.
4. Please once go through the manuscript. A few irrelevant words exist, such as (Fully characterized can be written as characterized also In the abstract section, Construction of molecules--in line 105 and many more like that).
5. Synthesized derivatives were reported as E-Geometrical isomers. Is there any mechanism adopted by authors to confirm the E-Geometrical form of the final derivatives (not the Z form)? If yes, please describe under results and discussion with reference.
Author Response
See the attached document

Reviewer 2 Report
Comments and Suggestions for Authors
3-amino-4-imino-3,5-dihydro-4H-chromeno[2,3-d]pyrimidine derivatives were synthesized and investigated of their in vitro antitumoral activities and in vivo toxicity by Karoui and co-workers. There are some points need to clarify;
-First of all, extensive editing of English required.
-No need to write compounds' name everytime. The authors can write just like "compound 1".
- What is the meaning of ref. 19? This ref. should be removed.
-The synthetic protocol of the target compouds is already known. Is it necessary to mention retrosynthetic strategy?
-The spectra (NMR and HRMS) of the compounds should be added as supplementary file.
- In antiproliferative assay, the author have used roscovitin and doxoruibicin but in zebrafish model sorafenib and erlotinib were used as standart. Why they were chosen differently?
- Embryotoxicity and angiogenesis Studies are in vitro or in vivo?
(Section 2.6)
- Is there any specific reason not to apply MW for the synthesis of compounds 8.
- Do the authors think about optimization studies for MW method to access excellent yields?
- In materials part, it would be more accurate to write only the synthesized compound not like ... from ..... and .....
Comments on the Quality of English Language
There are many grammatical and spelling errors. The manuscript should be checked carefully throughout.
Author Response
See the attached document

Reviewer 3 Report
Comments and Suggestions for Authors
The paper presents the synthesis, physicochemical and pharmacokinetical properties, antiproliferative activity and toxicity tests in zebrafish of new N-benzylidene derivatives of 3-amino-4-imino-3,5-dihydro-4H-chromeno[2,3-d]pyrimidines.
The molecules are new and fully characterized by 1H and 13C NMR, FTIR and HRMS. The chemistry strategy is interesting and the reaction yields are acceptable. The work is well planned and described. The experimental procedures are described comprehensively. The results are interesting. The manuscript is written clearly and concisely.
In my opinion, the paper is worth studying and the manuscript contains enough original and interesting material to enrich the research so far. It requires some modifications before being published which I have outlined in the comments below:
Minor corrections:
Lines 41-42, 89, 341, 351, 383, 385, 388, 390, 394, 397, 402, 417, 418, 825, etc : instead of “Zebrafish danio rerio” should be: “zebrafish Danio rerio”
Line 48: there is: figure 1, should be: Figure 1
Cytotoxicity studies: selectivity indices for the compounds and standards drugs should be calculated and placed in the table.
Supplementary Material is not attached.
Lines 357-368: these are not research results - this paragraph should be placed to “Materials and Methods”.
Figure 6 is completely illegible and does not allow for the assessment of the toxicity of the compound 10h and standard drugs towards zebrafish embryos and larvae.
Author Response
See the attached document

Reviewer 4 Report
Comments and Suggestions for Authors
In this manuscript, Sirine Karoui and colleagues presented the synthesis, ADME prediction, and biological evaluations of a series of new molecules featuring the 4H-chromeno[2,3-d]pyrimidines scaffold. 12 new molecules have been synthesized in a five-step synthetic route including two microwave-assisted reactions to validate the synthetic conditions and yield. Subsequently, all the new molecules are evaluated in both tumor cell lines and normal cells. Notably, two molecules, 10h and 10i demonstrated promising cytotoxicity against tumor cell lines in contrast to normal fibroblasts. Following the biological evaluations, the physiochemical characteristics and ADME properties of 10h and 10i were then assessed using in silico prediction tools. While in silico approaches are convenient, they cannot entirely replace actual experiments. Therefore, although the pharmacokinetics study was acceptable, it seems less convincing. Moreover, the in vivo zebrafish models provided additional confirmation that molecule 10h had no angiogenic effects during embryo development.
Figure 5 improvement:
It is difficult to interpret the inhibition dose-dependent manner (10-50uM) in Fig 5B. Therefore, if Fig 5B does not significantly impact the conclusions, it is advisable to remove it from the manuscript to avoid confusion.
Reference cited improvement:
In the introduction section, authors should reference the first published research article for HA14-1: Wang et al., Proc Natl Acad Sci USA. 2000 Jun 20;97(13):7124-9. Doi:10.1073/pnas.97.13.7124.
Several typos were identified in the manuscript:
1. Line 121, “Retrosynthetic Synthetic route” should be corrected to “Synthetic route”.
2. Line 401, “3.3 uM dose” should be “0.33 uM dose”.
Author Response
See the attached document

Round 2
Reviewer 2 Report
Comments and Suggestions for Authors
The authors made the necessary corrections and additions. The paper can be accepted in its current form.